# Review of Atom Chips for Absolute Gravity Sensors

**DOI:** 10.3390/s23115089

**Published:** 2023-05-26

**Authors:** Dezhao Li, Wenfeng He, Shengnan Shi, Bin Wu, Yuhua Xiao, Qiang Lin, Long Li

**Affiliations:** 1Zhejiang Provincial Key Laboratory of Quantum Precision Measurement, Collaborative Innovation Center for Information Technology in Biological and Medical Physics, College of Science, Zhejiang University of Technology, Hangzhou 310023, China; dezli@zjut.edu.cn (D.L.);; 2Science and Technology on Vacuum Technology and Physics Laboratory, Lanzhou Institute of Physics, Lanzhou 730000, China; 3Department of Aeronautics and Astronautics, Fudan University, Shanghai 200433, China

**Keywords:** atom chips, absolute gravity sensor, cold atom, microfabrication technologies

## Abstract

As a powerful tool in scientific research and industrial technologies, the cold atom absolute gravity sensor (CAGS) based on cold atom interferometry has been proven to be the most promising new generation high-precision absolute gravity sensor. However, large size, heavy weight, and high–power consumption are still the main restriction factors of CAGS being applied for practical applications on mobile platforms. Combined with cold atom chips, it is possible to drastically reduce the complexity, weight, and size of CAGS. In this review, we started from the basic theory of atom chips to chart a clear development path to related technologies. Several related technologies including micro-magnetic traps, micro magneto–optical traps, material selection, fabrication, and packaging methods have been discussed. This review gives an overview of the current developments in a variety of cold atom chips, and some actual CAGS systems based on atom chips are also discussed. We summarize by listing some of the challenges and possible directions for further development in this area.

## 1. Introduction

The absolute gravity sensor has been proven to be a powerful tool in scientific research and industrial technologies. In scientific research areas, it is considered to be a powerful tool to detect weak equivalence principle (WEP) violations [1,2], which can directly imply the equivalence between inertial and gravitational mass. Moreover, as a powerful tool, the absolute gravity sensor also has been widely applied to searching for dark energy [3,4], gravitational waves [5,6], determining the Newtonian gravitational constant *G* [7,8], short–range forces [9,10] and so on. On the other hand, this sensor has been widely applied in industrial areas including inertial navigation [11,12], archeology [13,14], monitoring earthquakes [15,16], groundwater [17,18], resource exploration [19,20], and many other applications.

Cold atom absolute gravity sensors (CAGSs) based on cold atom interferometry have been proven to be the most promising new generation absolute gravity sensor [21,22]. The prototype of the cold atom gravimeter was demonstrated by Steven Chu’s group at Stanford University, in 1991 [23]. The research of CAGSs has made great progress over the last 30 years. Different from the classical optical absolute gravity sensor using mechanical mass as a free-falling object, CAGSs use the cold atom cluster as the free–falling object to sense the gravity with the advantages of better stability, a long service life [24,25] and a high sampling rate [25]. The sensitivity and accuracy of CAGSs have exceeded the performance of the traditional optical absolute gravimeter of FG-5 [24,25]. As reported, the uncertainty of a CAGS was demonstrated to be high, up to 1.3 μGal (1 μGal = 10^−8^ m/s^2^) [26], while the sensitivity of a CAGS can also be high, up to $4.2\;\mu Gal/\sqrt{Hz}$ [27]. There are many research groups involved in this research area and some reported representative works of CAGSs are summarized in Table 1. The early research of CAGSs mainly focused on improving sensitivity and stability. Over the past decade, researchers gradually shifted their attention to miniaturization and applications of CAGSs. Portable CAGSs have been demonstrated to be applicable on various mobile platforms, including vehicles [28,29], shipboard [30,31], airborne [32,33], and even in aerospace [18,34,35]. However, weight, power consumption, and noise in a dynamic environment are still the main restriction factors of CAGSs being applied for practical applications, especially on mobile platforms [16,19,34,36,37]. Many different strategies have been proposed to solve these problems, and using atom chips is one of the most promising methods, since they can significantly reduce the complexity, weight, and size of CAGSs.

An atom chip was originally proposed to miniaturize setups of cold atom systems using some mature technologies in integrated circuit areas [43,44]. The first atom chip was reported in 1999 [45], and only a few years later the Bose–Einstein condensation (BEC) was realized with an atom chip [46,47]. Subsequently, the development of atom chips has brought many significant changes and surprises to cold atom sensors including CAGSs, as in Figure 1.

In this review, the recent advancements in CAGSs with atom chips have been reviewed from different technical perspectives, including fundamental theory, different proposed designs, types of substrate materials, and various fabrication technologies. Moreover, some realized CAGS systems based on atom chips are also discussed, and some possible strategies are also mentioned for further improvement to realize a chip–scale CAGS with all components fully integrated onto a chip.

## 2. Basic Theory of CAGS and Magneto–Optical Trap (MOT)

The working principle of a CAGS is based on the cold atom, and the preparation of a cold atom cloud (with a temperature below 10 μK) is an essential process for a CAGS. Following the atomic cooling techniques proposed by several Nobel laureates based on the Doppler effect [51,52], neutral atoms can be trapped by a magneto–optical trap (MOT) first, and then the trap uses polarization gradient cooling to further reduce the temperature of the atom cloud, so that the coherence of atoms can be further improved. With a well–prepared cold atom cloud, a three–pulse sequence (π/2–π–π/2) will be applied to split, redirect and recombine the matter–wave of cold atom clouds as a typical Mach-Zehnder interferometer as in Figure 2. The accumulated phase difference includes the free evolution phase determined by Feynman’s path integral approach, and the laser–atom interaction with gravity information contained. Through careful calculation, the gravity acceleration can be extracted from the interference fringes, obtained by scanning one of these variables, such as the chirp rate of Raman optical frequency [51].

As an important component of CAGSs, the magneto–optical trap (MOT) was originally proposed and realized by Raab et al., as the most widely used method for cold atom cloud preparation [53,54,55]. The fundamental work principle of the three–dimensional MOT was realized by three pairs of circularly polarized red–detuned lasers with a static gradient magnetic field, as in Figure 3a. The static gradient magnetic field is usually around 10 G/cm, traditionally provided by a pair of anti–Helmholtz coils, and the magnetic field strength is zero at the intersection of three pairs of lasers [56]. The energy level of these atoms in the MOT will split due to the Zeeman effect, and one possible configuration of the energy level structure is as shown in Figure 3b. We can set the total angular momentum of the ground state as J=0, and J=1 for the excited state, which contains three magneton energy levels $m_{f}=0,\pm 1$ generated by Zeeman splitting in the magnetic field. These magneton energy levels degenerate at the center of the magnetic field. Taking *Z* direction as an example, as in Figure 3b, two lasers collide these atoms from opposite directions. The frequency of these lasers is $w_{Laser}$, the detuning frequency of which is relative to atomic resonance frequency at zero magnetic fields, or $\delta_{0}$. In addition, the polarization directions of these two lasers are σ+ and σ−. Due to the transition selection rules, the detuning of σ+ and σ− lasers are $\delta^{+}$ and $\delta^{-}$, respectively. Assuming the initial position of an atom is *Z*_0_ (*Z*_0_ > 0), then $\delta^{+}>\delta^{-}$ and atoms will absorb more photons from σ− lasers, resulting in a negative centripetal force to atoms. Similarly, the atoms will gain a positive centripetal force if *Z*_0_ < 0, and all atoms will feel the radiation force from lasers pointing to the center, becoming trapped in the center as in Figure 3b.

The traditional MOT is usually constructed by the combination of the magnetic coil and laser system, making the experimental system complicated and very heavy. Recently, with the development of integrated circuits and micro/nanofabrication processing, various designs have been proposed to simplify this system with different types of atom chips. The atom chips are usually fabricated with surface micro/nanostructures to accurately control the magnetic, electric, or light field. Using these atom chips, the confinement and coherent manipulation of cold atoms can be realized on a small scale under low power consumption conditions.

## 3. Micro Magnetic Traps Design on Chips

The classical magnetic traps are generated by winding electric coils with large volume and high–power consumption [57,58]. To overcome these shortcomings, the fundamental concept proposed by some pioneer scientists is combing an atom chip with a homogenous bias field. The current–carrying wire on atom chips can be directly fabricated with nanofabrication technology [59]. In addition, by using a substrate with high heat dissipation performance, atom chips can carry enough current density to form different static and dynamic traps, which can be applied to capture, trap, and transfer neutral atoms. There are various kinds of micromagnetic traps with atom chips that have been proposed including basic wire traps [43,60], micro Ioffe–Pritchard–type magnetic traps (MIP traps) [44,61], and combined magnetic traps [62,63].

Atom chips with basic wire traps are the earliest proposed and fabricated devices [43], working with an external uniform magnetic field. There are mainly three different types of structures, including single wire, Z, and U–shaped micromagnetic traps, as described in Figure 4. The side guide trap was realized with a straight current–carrying wire in a homogenous bias magnetic field as shown in Figure 4a. For this trap, the depth can be determined by the bias field and the gradient is inversely proportional to the wire current. Using this straight wire, a modified trap with wire displaced from the quadrupole axis was applied to get a BEC to condense on an atom chip [64,65]. By bending a wire into a “U” shape, a three–dimensional quadrupole filed with a zero magnetic field point can be formed, as shown in Figure 4b. With this structure, a three–dimensional quadrupole trap can be realized and a rotation of the bias field displaces the trap minimum, but the field always vanishes. For a “Z” shape wire, we can get an Ioffe–Pritchard–type trap without zero magnetic fields, as shown in Figure 4c. In all, the potentials for the “U” and “Z” shapes are similar, but the fields of finite length near the central bar are varied.

An Ioffe–Pritchard–type trap, which is a kind of magnetic trap without zero magnetic fields consisting of Helmholtz coils and four straight wires, were originally proposed by Pritchard et al., in 1983 [67]. Based on the working principle of the Ioffe–Pritchard–type trap, Weinstein and Libbrecht put forward several planar wire designs for atom chips in 1995 [44], shown in Figure 5. One of the essential planar structures to an analog of the traditional Ioffe–Pritchard trap was proposed with two loops and four Ioffe bars as shown in Figure 5a. Based on the essential structure, one of the loops can be replaced with a bias field to get a multipole trap, as shown in Figure 5b. Other adjustments have also been discussed and these structures are considered to be very suitable for using superconducting materials to construct traps with magnetic–field gradients greater than $5\times 10^{5}\;G/cm$.

Although these mentioned wire designs were seldom directly used for real applications in atomic sensors, these designs provided a basic reference point for the subsequent proposed combined magnetic traps. The combined magnetic trap is composed of two single–type micromagnetic traps, according to different requirements, as shown in Figure 6. In 2009, Shengwang Du and Eun Oh proposed a simple three–wire–based magnetic trap as shown in Figure 6a, and the potential depth and frequencies can be controlled independently by tuning the operational current of the wire and external bias fields [62]. With this proposed magnetic trap configuration, the authors provided a possible method to obtain Bose–Einstein condensation without the traditional forced evaporation system. Subsequently, researchers discussed a combined wire pattern of U and H–wire traps [68,69] as shown in Figure 6b, and different types of magnetic traps can be realized by selecting different working wires based on this structure. Recently, more combined wire patterns have been designed, which can provide more flexible magnetic traps and lay a foundation for the application of atomic chips for CAGSs. Meanwhile, magnetic design is the essential step for a CAGS design and some novel designs are still widely discussed. Other than magnetic shapes with different kinds of wire structures, the heat–caused high temperature is also a critical problem for wire trap design, which has also been widely discussed in many other fields [70,71,72].

The characteristics and usage of these mentioned magnetic traps are varied. For practical applications, a suitable wire pattern needs to be designed with a whole system of CAGSs. From the perspective of atomic capture and trapping, “U” type wire is mainly used for micro–MOT combined with optical traps while “Z” type wire can be used for atom traps without a zero–point magnetic field. Moreover, the noise of the atom chip is mainly caused by current noise, electric field fluctuation, spontaneous emission, Casimir-Polder (C–P) force, Van der Waals force, and surface adsorption, which can shorten coherence length and coherence time. To maintain or even enhance the signal-to-noise ratio of CAGSs, one possible strategy is to lower the atom temperature and thus improve the coherence, and another strategy is to adopt a wire waveguide to split the atoms in time to get longer interaction time [73].

## 4. Magneto–Optical Trap Realized by Various Atom Chips

Integration of an optical trap and magnetic trap to form a magneto–optical trap for atom chips was also widely discussed in previous research. Several strategies have been successfully implemented including a mirror–MOT [74,75], a magneto–optical trap with a pyramid structure [76,77], and a magneto-optic trap with a grating structure [78,79]. The mirror–MOT scheme was proposed to apply a metal film with a certain thickness on the substrate, which can provide a mirror surface with high reflectance, to further simplify the experiment from the original 6 laser beams to 4 laser beams [45,80]. With a mirror surface structure, the substrate of the atom chip can be non–transparent, and around $5\times 10^{6}$ atoms were realized in a MOT with a distance from a surface larger than 1 mm [45]. For high–precision gravimeters, the reflectance of the atom chip should be higher than λ/10 without influencing the wave front quality too heavily [49].

Furthermore, some scientists discussed a type of magneto–optical trap realized with an integrated MOT, with U and Z–wire traps as shown in Figure 7a [80]. In this reported structure, the atom chip was used as a mirror and the U and Z–wire traps were realized with copper pieces incorporated with a macro–ceramic block. Through combing the mirror atom chip and wire magnetic traps, more than $3\times 10^{8}$ rubidium (^87^Rb) atoms were obtained, which is a very considerable number for the realization of CAGSs.

The pyramid MOTs were realized by the integration of copper wire and pyramid structures, which were fabricated using anisotropic wet etching followed by polishing in an ICP etch, as shown in Figure 7b [81]. With this kind of structure, around 2000 atoms were trapped in one single MOT, which is not enough for a CAGS application. However, this pyramid structure MOT just needs one single laser beam to trap atoms, which can be applied to simplify the MOT structure [83]. Inspired by this design, a CAGS was realized using a large–scale pyramidal structure with a sensitivity of $1.7\times 10^{-6}m/s^{2}$ at one second [84]. Moreover, this typical pyramid MOT has been explored for other applications including gravity gradiometers [85], multi-axis atom interferometry [86] optical lattice clocks [87], and so on.

Other than reflective light field regulation methods mentioned above, some researchers proposed grating MOT chip structures based on this strategy to regulate light fields with different grating structures using the working principle described in Figure 7c [82]. Compared with micro–pyramid MOTs previously reported, this kind of grating type structure can provide a large atom number over $1\times 10^{7}/{cm}^{3}$. The number of trapped atoms in this structure was determined by the beam overlap volume, laser frequency and intensity. Based on this reported structure, a flight capable CAGS was realized recently [88]. What’s more, grating MOTs have also been demonstrated for compact quantum sensors where the distance of the ultracold atom can be more than a centimeter away from the grating plane without being influenced by the input laser beam power [89].

With these proposed chip–scale MOTs, some researchers have made progress in the development of miniaturized CAGS, and new chip–scale MOT schemes are still being explored [90,91,92]. Two attractive optional chip–scale MOT designs are presented in Figure 8. A novel MOT integrated with a metasurface optical chip, which was designed to diffract single incident light into five beams to simplify the MOT system, was proposed as in Figure 8a, and ~10^7^ atoms were captured with ~35 μK. This performance is suitable for CAGS, which can be explored in real applications in the future.

Another interesting proposed chip–scale MOT structure was realized by combing an optical grating chip and a flat magnetic coil chip [92]. In this structure, the optical grating chip can simplify the conventional six-beam configuration to a single laser beam, and the flat coil chip can simplify the anti–Helmholtz coils into a cylindrical geometric shape. With this structure, around 10^4^ rubidium (^87^Rb) cold atoms were captured above the chip surface. These novel MOT structures provide new strategies for further applications and more chip–scale MOTs still need to be explored.

To further improve the portability of CAGSs, micro–fabricated lenses [93,94,95] ultra-high vacuum cells [96,97,98], and even micro ion pumps with a size of around 2.5 cm^3^ [99,100,101] are already under discussion.

## 5. Material and Fabrication Technologies of Different Atom Chips

The original concept and manufacturing technology of atom chips were inspired by the development of integrated circuits (IC) and micro–electro–mechanical system (MEMS) technologies [59,102,103,104]. The substrate material has a significant effect on the fabrication and operation mode of atom chips. Many different kinds of materials have been applied as substrates for atom chips, and several typical materials are summarized in Table 2.

The substrate of the atom chip is applied to support the wire patterns. It should meet some basic requirements, such as high electrical insulating quality, high thermal conductivity, and feasible mechanical properties for fabrication and packaging. At present, the most commonly used material in experiments is a mono–crystalline silicon wafer, which is generally about 500 µm thick and easy to process with traditional IC fabrication technologies [68]. Due to the mature processing technology, the roughness of the silicon wafer can be less than 5 nm and the thermal conductivity of the silicon wafer is around 148 W/(m·K) at room temperature. In terms of electrical insulation, usually, a layer of silicon oxide (Si) is plated on the surface of the substrate by thermal oxidation or plasma-enhanced chemical vapor deposition (PECVD) to increase the electrical resistivity of the substrate surface.

Other than silicon, researchers also attempt to explore other materials to create a more efficient design for atom chips. With better thermal conductivity, Silicon carbide (SiC) has also been applied as the substrate to obtain a higher carrying current for a deeper magnetic trap [105,107]. In addition, thanks to its transparent property against a laser beam of 780 nm, the MOT combined with a SiC atom chip is easy to realize based on the conventional MOT system. In the previous report, around 10^8^ rubidium (^87^Rb) atoms have been captured, which is very considerable for CAGS applications. Aluminum nitride (AlN), with high thermal conductivity and intrinsically high electrical resistance, was also applied as a substrate material for atom chips [106,108]. Due to its unique property, the maximal carrying current with AlN was demonstrably high, up to 16 A. An atom chip with a glass of BK7 substrate was also discussed due to its transparent property and the maximum carrying current was around 2 A. These substrate materials of atom chips provide different solutions for the design and integration usage of compact MOTs applied to CAGS. The material for the wiring pattern also needs to meet some specific requirements, including high adhesion properties with a substrate, high current–carrying capacity, and thermal stability [109,110,111]. To capture atoms with atom chips, a load current of over 1 A is usually required to obtain a sufficiently large magnetic gradient. Copper (Cu) [68,80,112,113,114] and gold (Au) [82,115,116], with high electrical conductivity and high reflectance were reported as wire materials. Comparing these two materials, the property of Au is stable, but the electrical conductivity of Cu is higher and they can both be applied to atom chips according to different requirements. To improve the performance and stability of atom chips, new materials with excellent electrical and thermal properties still need to be explored.

The electrocaloric effect caused by the atom chip may influence the performance of CAGSs. On one hand, thermal noise should be considered, but on the other hand, heat dissipation may affect the number and temperature of atoms. Related research should be conducted, including the electric control and thermal transfer designs of atom chips. For optimization of thermal transfer designs, more research should be conducted, including wire pattern designs, substrate material selections, and so on.

Since there are many different reported structures of atom chips, there is no unified fabrication process. For wire patterns on the substrate, there are two common fabrication strategies: (1) using a thick adhesive photoresist with lithography to prepare the wiring pattern, growing the metal, and finally stripping the photoresist to obtain the final structure [117,118,119]; or (2) covering a metal layer on the substrate, and then using lithography to form the wiring pattern, and etching the final structure [120,121]. The current-carrying metal layer can be prepared via electroplating deposition, magnetron sputtering, or electron beam evaporation. Each of these technologies has its advantages and disadvantages. Electroplating deposition can rapidly grow metal layers, but due to the uneven internal structure, the surface is rough and easily oxidized. A thin metal film (<5 μm) can be made via magnetron sputtering or electron beam evaporation technology, which is very smooth and compact, and the adhesion properties between a metal layer and the substrate are strong. For pyramid–type atom chips, some micro/nano–electro–mechanical system (M/NEMS) fabrication technologies have been applied [81,122,123] including anisotropic wet etch with a KOH solution, inductively coupled plasma (ICP) etches, and so on. Usually, the design and fabrication technologies of atom chips should be considered simultaneously. For some novel structures, other delicate fabrication technologies still need further exploration.

## 6. Package Methods of Atom Chips for CAGSs Application

Due to the working principle that a cold atom cloud needs a falling distance to sense the gravity value in an ultrahigh vacuum environment, special vacuum package technologies should be considered to develop a CAGS system with these atom chips. The main problem with the package is determining how to export the electrical wire of the atom chip from the vacuum chamber.

There are several strategies to achieve ultra–high vacuum packaging of atom chips in a vacuum chamber. The pioneer package method, using the traditional feed–through technique to input current for the atom chip, was to package the atom chip directly into a vacuum chamber with a vacuum flange [124], as shown in Figure 9a. This traditional packaging method has high stability but the package volume is complex and relatively large. For atom chips based on a silicon wafer, the wafer can be directly bonded with a glass vacuum chamber via an anodic bonding method as shown in Figure 9b. In addition, another technique of through silicon via (STV) was applied to input current directly to the chip from the back of the silicon wafer, as shown in Figure 9c. This method can provide an ultrahigh vacuum ($9.5\times 10^{-10}$ Torr) environment for CAGS application. Moreover, the volume was reduced with a simplified system. Other kinds of vacuum packing methods were also discussed, such as sealing the chamber with high–temperature glue, glass solder sealing and indium wire welding, and so on. A CAGS system requires different kinds of vacuum packaging methods, and suitable packaging methods can reduce the volume and weight of the system.

For the combination of an atomic chip and a macro scale vacuum chamber, usually researchers apply one single high vacuum chamber to simplify the manipulation measurement process. However, for a chip–scale vacuum chamber (with a size of around 1 cm^3^), the dual–chamber design can reduce the influence of atom generation on the measurement performance.

## 7. Different CAGSs Realized Using Atom Chips

Although atom chips have been proposed for several decades, only a few CAGS systems have been demonstrated. In 2016, Abend et al. from the University of Hannover, Germany reported a BEC Fountain gravimeter based on atomic chips [49]. They applied the atom chip to prepare the BEC and used this chip as a mirror to form the pulse lattice to drive the Bloch oscillation and double Bragg diffraction to form the Mach–Zehnder interferometer. The intrinsic sensitivity of this system is around $5.3\;ng/\sqrt{Hz}$ in a volume of around 1 cm^3^ and makes miniaturized sub–µ Gal (1 µGal = 10 nm/s^2^) CAGS feasible. In this work, the authors discussed a new scheme to extend interferometer time by employing BEC interferometry with a relaunch in a compact volume, which provides a possible prototype for miniaturized CAGSs.

Limited by microfabrication and small–size light source technologies, there are still no all–on–a–chip CAGSs (with a chip-scale vacuum chamber, chip–scale laser source, and so on) demonstrated at present. However, with the development of atom chip technologies, researchers realize that a chip–based or even a chip–scale CAGS has great advantages in terms of miniaturization applications, such as in space and elsewhere. The research efforts in any related work on cold atom chips may provide an important contribution to the realization of CAGSs in the future.

## 8. Conclusions

In this review, recent progress in atom chip-based CAGSs has been presented along with a basic theory of micro-MOTs, micromagnetic traps and MOTs realized with atom chips, material and fabrication technologies, package methods, as well as prototype CAGS realized with atom chips. The content was organized following the development of atom chips and their applications in CAGSs.

As the essential component of CAGS, the working mechanism of MOTs provide clear guidance as to the design of atom chips. Firstly, micromagnetic traps generated with electrical wire on chips can reduce the volume of MOTs. Although the idea is very straight forward and many wire patterns have been demonstrated, better solutions for atom-scale magnetic traps for CAGSs are still needed. Combined with on-chip light field regulation, various MOTs, including mirror MOTs, pyramid MOTs, grating MOTs, MOTs with dielectric metasurfaces and Planar integrated MOTs have been presented. However, determining how to apply these MOTs to miniaturized CAGSs is still under discussion. To realize applicable atom chips for CAGSs, different materials applied to substrates and wires have been discussed. Furthermore, to integrate the atom chips into a CAGSs, different ultrahigh vacuum packaging methods have to be considered. With some of these methods, CAGSs with atom chip integration have been demonstrated, but more efforts are needed to develop a smaller CAGS for movable platform applications. Still furthermore, how the wiring pattern, heat management, and vacuum package will influence the performance of CAGSs needs to be researched more thoroughly for each of the different measurement systems.

Further development of chip-scale CAGSs can be conducted by investigating a method to realize a longer interference time. One possible direction is to integrate a cold atom chip with an optical lattice to develop a long interference time for gravity measurement. High power on-chip light sources used for cold atom trapping are also an important limitation. Ultrahigh chip-scale vacuum packaging on a chip has not been applied to CAGSs, which also warrants further investigation. With the development of related technologies, a smaller and better miniaturized CAGS should be realized in the future.

## Figures and Tables

**Figure 1 sensors-23-05089-f001:**
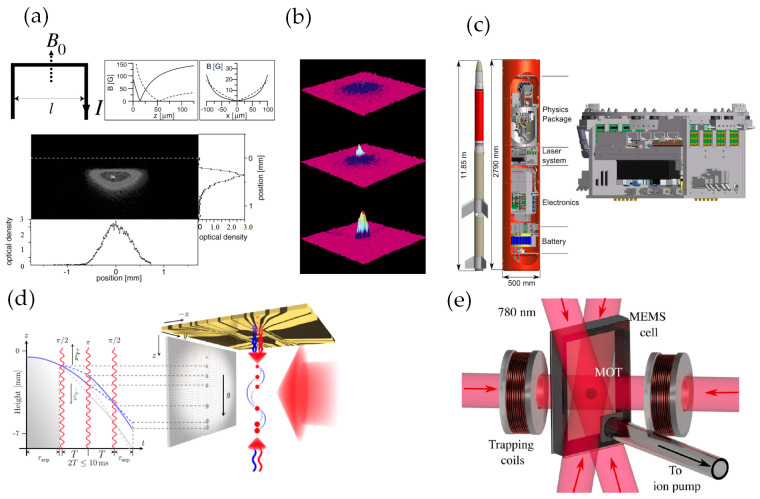
(**a**) Wire layout of an atom chip for quadrupole potential with generated magnetic potentials and an image of a cold atom cloud (photo reproduced with permission from American Physical Society) [45]; (**b**) Time–of–flight absorption images of BEC generated with an atom chip (photo reproduced with permission from Springer Nature) [47]; (**c**) the Matter–wave Interferometry in Microgravity (MAIUS) rocket (photo reproduced with permission from Springer Nature) [48]; (**d**) Atom–chip based gravimeter and space-time trajectories of a BEC (photo reproduced with permission from American Physical Society) [49]. (**e**) Image of a chip–scale MOT (photo reproduced with permission from AIP publishing) [50].

**Figure 2 sensors-23-05089-f002:**
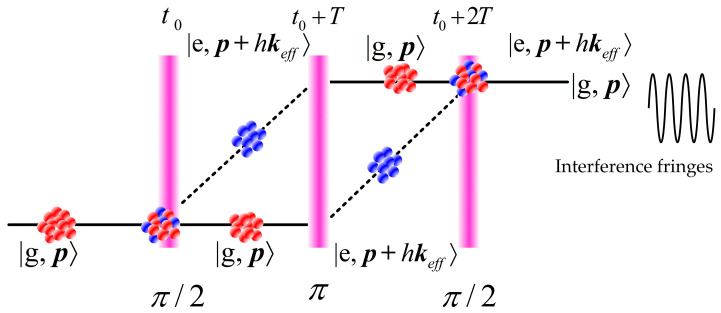
Pulse sequence process of an atom interferometer.

**Figure 3 sensors-23-05089-f003:**
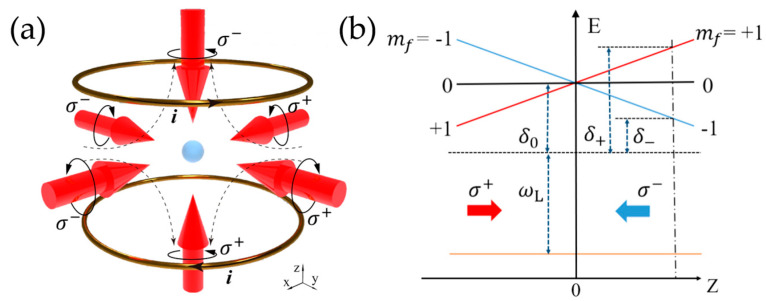
(**a**) Schematic diagram of a three–dimensional magneto–optical trap. (**b**) Atomic energy level splitting and trapping principle.

**Figure 4 sensors-23-05089-f004:**
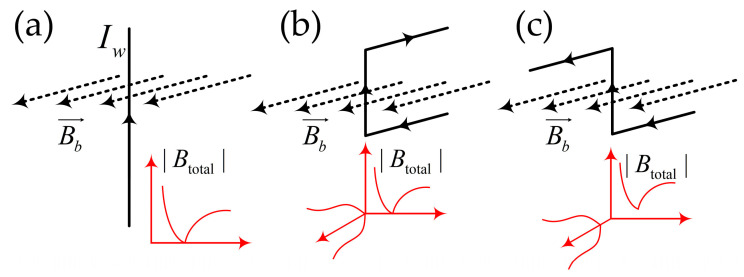
Three different basic traps (**a**) a side guide trap and its potential shape, (**b**) a U–shaped micromagnetic trap and its potential shape, (**c**) a Z–shaped micromagnetic trap and its potential shape, reproduced from [66] with permission from Springer Nature.

**Figure 5 sensors-23-05089-f005:**
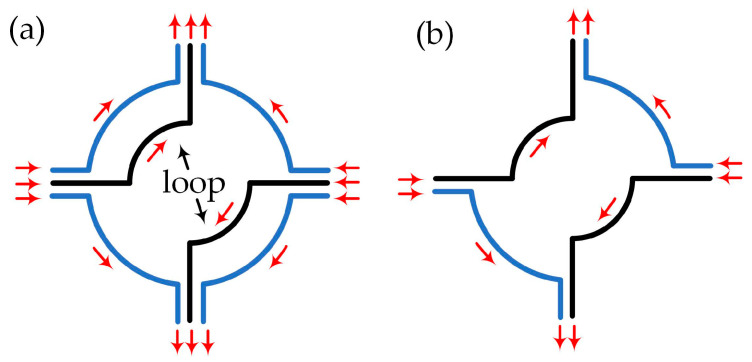
A Planar Ioffe–Pritchard–type micro–magnetic trap with (**a**) three concentric half loops, (**b**) two half loops with an external bias field, reproduced from [44] with permission from the American Physical Society, and the arrow represent the direction of the current.

**Figure 6 sensors-23-05089-f006:**
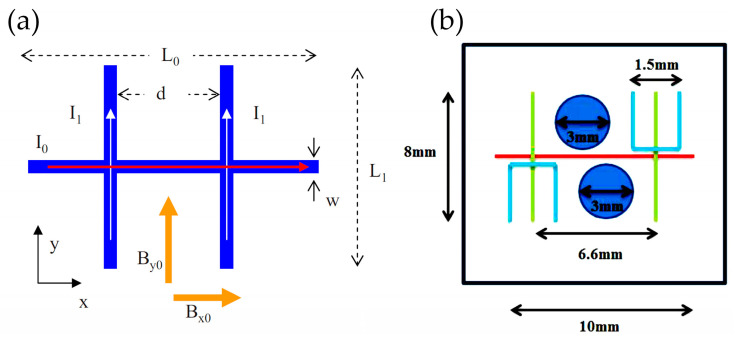
(**a**) A schematic of an “H” type wire trap design accompanied with external bias magnetic fields reproduced from [62] with permission from American Physical Society, and the arrow represent the direction of the current. (**b**) A schematic of a combination U and H wire–pattern trap reproduced from [69] with permission from MDPI AG.

**Figure 7 sensors-23-05089-f007:**
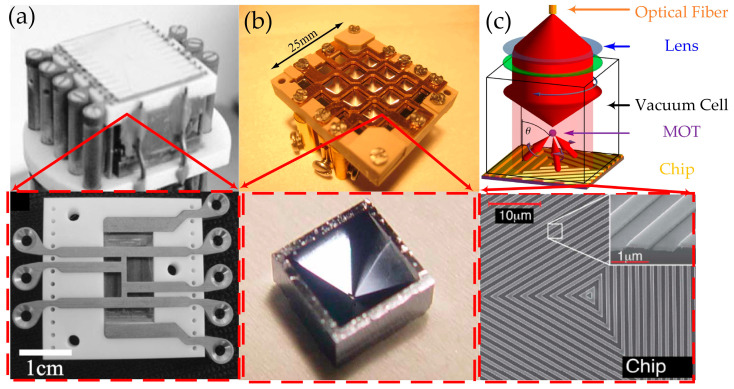
(**a**) A mirror–type magneto–optical trap combined with U and Z wire magnetic traps [80] reproduced with permission from American Physical Society, (**b**) Silicon pyramid MOTs with an enlarged picture of the pyramid structure [81] with permission from Optica Publishing Group under the terms of the Open Access Publishing Agreement, (**c**) Grating chip MOT with scanning electron microscope (SEM) fabricated grating structure [82] with permission from Springer Nature.

**Figure 8 sensors-23-05089-f008:**
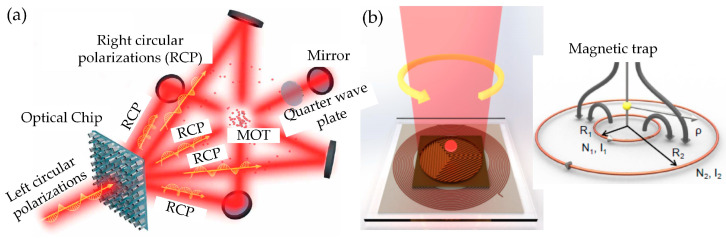
(**a**) A cold atom MOT with the dielectric metasurface optical chip [91] reproduced with permission from AAAS Publication, (**b**) A planar–integrated MOT through combing an optical grating chip (**left**) and magnetic coil chip (**right**) [92] reproduced with permission from the American Physical Society.

**Figure 9 sensors-23-05089-f009:**
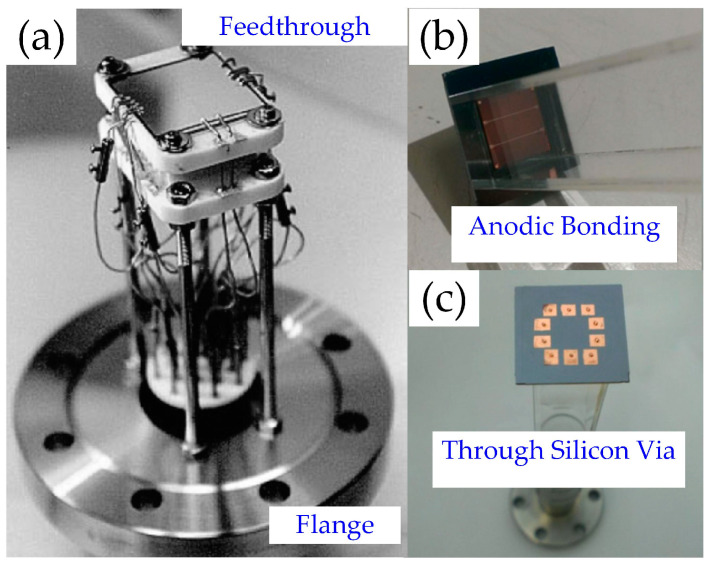
(**a**) The atom chip with feedthrough manufactured before it is packaged into a vacuum chamber with flange, reproduced [124] with permission from American Physical Society. (**b**) The atom chip is packaged with a vacuum chamber using an anodic bonding technique. (**c**) Current input to the atom chip based on the TSV technique, reproduced [125] with permission from IOP publishing.

**Table 1 sensors-23-05089-t001:** State–of–the–art of gravity sensors from different research groups as reported.

Research Group	$\mathrm{Sensitivity}\;(\frac{\mu Gal}{\sqrt{Hz}})$	Uncertainty (μGal)	Ref.
Stanford Uni.	8	3.0	[38,39,40]
Humboldt Uni.	9.6	3.2	[19]
SYRTE	5.6	1.3	[25,26,41]
HUST	4.2	3.0	[27,42]

**Table 2 sensors-23-05089-t002:** Different properties of the applied substrate materials.

Substrate Material	Chip Function	Thickness (μm)	Thermal Conductivity (Wm^−1^K^−1^)	Transparent or Not	Ref.
Silicon (Si)	Double layer micro-magnetic trap	500	148	No	[68]
Silicon carbide (SiC)	Micro-magnetic trap	414	>390	Yes	[105]
Aluminum nitride (AlN)	Micro-magnetic trap	600	128	No	[106]
Glass (BK7)	Micro-magnetic trap and Mirror	400	~1	Yes	[69]

## Data Availability

Not applicable.

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
