# Peer review of "Review of Atom Chips for Absolute Gravity Sensors"

_sensors, 2023, doi:10.3390/s23115089_

Round 1

Reviewer 1 Report

In the context of the development of new generation absolute gravity sensor, the Cold Atom Absolute Gravity Sensor appears as very promising, but needs a substantial work on reduction and improved power budget for on field application. This article presents an exhaustive list of what has been done in this context, particularly on the atom chip development for CAGS.

While the content and the reference of the work done worldwide are very well described, the English language used can and should be substantially improved. 

The reviewer recommends therefore the publication of this complete review on atom chips for CAGS after a proper English checking and writing, in order to get a proper scientific review article.

Author Response

We thank the reviewer for the careful examination, the constructive comments. To improve our manuscript, we have revised the manuscript accordingly to address the issues raised by the reviewer. All the revised and added parts have been marked in red in the manuscript. Please see the attachment to check the point-by-point responses to the reviewer’s comments.

Reviewer 2 Report

Dear Authors,

Please, see the attached report.

The Reviewer

Author Response

(The authors gave the same response as above.)

Author Response

(The authors gave the same response as above.)

Round 2

Reviewer 2 Report

Dear Authors,

Please, see the attached report.

Best regards,

The Reviewer

Author Response

(The authors gave the same response as above.)

Reviewer 3 Report

Page numbers of several references are omitted. Please give and complete them.

Author Response

We thank the reviewer for the careful examination, the constructive comments, and the positive evaluation. 
